# *Drosophila* as Model System to Study Ras-Mediated Oncogenesis: The Case of the Tensin Family of Proteins

**DOI:** 10.3390/genes14071502

**Published:** 2023-07-23

**Authors:** Ana Martínez-Abarca Millán, Jennifer Soler Beatty, Andrea Valencia Expósito, María D. Martín-Bermudo

**Affiliations:** Centro Andaluz de Biología del Desarrollo, Universidad Pablo de Olavide/CSIC/JA, Ctra Utrera Km1, 41013 Sevilla, Spain; amtezmi@gmail.com (A.M.-A.M.); beattyjs.94@gmail.com (J.S.B.); andrea.ve06@gmail.com (A.V.E.)

**Keywords:** oncogenic Ras, overgrowth, tensins, *Drosophila*

## Abstract

Oncogenic mutations in the small GTPase Ras contribute to ~30% of human cancers. However, tissue growth induced by oncogenic Ras is restrained by the induction of cellular senescence, and additional mutations are required to induce tumor progression. Therefore, identifying cooperating cancer genes is of paramount importance. Recently, the tensin family of focal adhesion proteins, TNS1-4, have emerged as regulators of carcinogenesis, yet their role in cancer appears somewhat controversial. Around 90% of human cancers are of epithelial origin. We have used the *Drosophila* wing imaginal disc epithelium as a model system to gain insight into the roles of two orthologs of human TNS2 and 4, *blistery* (*by*) and *PVRAP*, in epithelial cancer progression. We have generated null mutations in *PVRAP* and found that, as is the case for *by* and mammalian tensins, *PVRAP* mutants are viable. We have also found that elimination of either *PVRAP* or *by* potentiates *Ras^V12^*-mediated wing disc hyperplasia. Furthermore, our results have unraveled a mechanism by which tensins may limit Ras oncogenic capacity, the regulation of cell shape and growth. These results demonstrate that *Drosophila* tensins behave as suppressors of Ras-driven tissue hyperplasia, suggesting that the roles of tensins as modulators of cancer progression might be evolutionarily conserved.

## 1. Introduction

Cancer is the second leading cause of death worldwide [1]. Cancer is characterized by the uncontrolled growth and spread of cells leading to invasion of normal tissues. Extensive research over the last few years has revealed that cancer is a genetically complex and heterogenous disease with each tumor carrying mutations not in a single gene but in several genes [2,3]. Thus, the isolation of factors collaborating with tumor progression is crucial to understand tumor development and malignancy.

The *Drosophila* genome is 60% homologous to that of humans, and about 75% of genes responsible for human diseases have homologs in flies [4]. This, combined with a short generation time, low maintenance costs and powerful genetic tools, makes the fly an ideal model system to study cancer. About 90% of human cancers are of epithelial origin [5]. Epithelial tissues are composed of cells with an apico-basal polarity held together in sheets by specialized junctions. The apical face of the epithelial tissue is exposed to either the external environment or the body fluid, while the basal face is attached to a specialized extracellular matrix (ECM), called the basement membrane (BM). In this context, the primordium of the *Drosophila* wing, the larval wing imaginal disc epithelia, has been successfully used to study epithelial tumor progression and oncogenic cooperation (reviewed in [6]). The wing imaginal disc (henceforth, wing disc) is a monolayer epithelium that is limited apically by a squamous epithelium, the peripodial epithelium (PE), and basally by a BM. The formation of the wing starts during early embryonic development when about 30 cells are allocated to form the wing disc primordium. During larval life, the number of cells in the disc increases to about 50,000 [7]. The wing disc epithelium is often divided into four broad domains, the pouch, the hinge, the notum and the PE. The pouch and the hinge give rise to the wing and wing hinge, while the notum gives rise to the dorsal half of the body wall in the second thoracic segment, which includes the notum, the back of the thorax and the pleura (reviewed in [8]). The induction of groups of cells overexpressing mutated forms of *Ras* (*Ras^V12^*), present in 30% of human cancers [9], in the wing disc gives rise to hyperplastic growth [10]. This has been exploited, by several labs including ours, in screens to isolate new genes affecting the growth of cells overexpressing *Ras^V12^* [11].

To isolate new modulators of oncogenic Ras activity, we performed an RNAi screen and searched for genes that enhanced the wing disc *Ras^V12^* overgrowth phenotype when knocked down. One of the genes identified has been proposed to be the fly ortholog of human tensins 2 and 4 (TNS2, TNS4, FlyBase), which we have named *EGFRAP* [12]. Likewise, our preliminary results showed that downregulation of another fly ortholog of human TNS2 and 4 (FlyBase), *PVRAP*, a gene that lies adjacent to *EGFRAP*, increased the Ras^V12^ overgrowth and folding wing disc phenotype similarly to *EGFRAP*. Tensins are a family of focal adhesion proteins. The mammalian tensin family comprises four members (tensin 1-4, TNS1-4), which link the actin cytoskeleton to the cell membrane and are lost in many cancer cell lines [13,14]. Knockdown of human TNS2 and TNS4 increases tumorigenicity in several cancer lines [15]. Interestingly, the role of tensins as tumor suppressors has been linked to their ability to bind and regulate the main cell-ECM receptors, the integrins (reviewed in [14]), which are themselves involved in almost every step of cancer progression (reviewed in [16]). In this context, it is worth mentioning here that we have recently found that downregulation of integrins also enhanced the Ras^V12^ overgrowth and folding wing disc phenotype [17]. However, even though *blistery* (*by*) is the clearest *Drosophila* ortholog of vertebrate TNS4 [18] and has been proposed to provide a final strengthening of the integrin adhesion complex [19], it has not yet been implicated in tumorigenesis [18].

In this work, we used the wing disc to analyze the role of *PVRAP* and *by* in the progression of *Ras^V12^* epithelial tumor cells. We generated mutants in *PVRAP* and found that, as is the case for mutations in *by*, *PVRAP* mutants are viable, strongly suggesting that *PVRAP* function is not required for development. In addition, we found that downregulation of either *PVRAP* or *by* enhances *Ras^V12^*-mediated tissue hyperplasia. Our results also unravel a new mechanism by which tensins could modulate the oncogenic capacity of *Ras^V12^*, the regulation of cell shape and growth. These results demonstrate that in *Drosophila*, as is the case in some human cancer cell lines, tensins modulate the metastatic capacity of *Ras^V12^*-dependent epithelial tumor cells, suggesting that the role of the tensin family of proteins as a suppressor of tumor metastasis might be evolutionarily conserved.

## 2. Materials and Methods

### 2.1. Fly Strains

The following fly strains were used: *UAS-by^RNAi^*, *UAS-PVRAP^RNAi^* (Vienna Stock Center, Vienna, Austria); *UAS-Ras^V12^* [20]; *apGal4-UASGFP* (Bloomington Drosophila Stock Center, Bloomington, IN, USA). The two *PVRAP* mutants *PVRAP^1^* and *PVRAP^2^* were isolated using CRISPR in this study (see below). Flies were raised at 25 °C.

### 2.2. Isolation of PVRAP Mutants Using CRISPR/Cas9

To generate null alleles, we designed two sgRNAs against sequences located in the first exon, close to the ATG. The following sequences were used:

sgRNA1: GTCGCCAGCAGCACAATAATAGC;

sgRNA2: AAACGCTATTATTGTGCTGCTGG;

sgRNA3: GTCGGAATTGGAATTGTCCGCCG;

sgRNA4: AAACCGGCGGACAATTCCAATTC.

The sgRNAs were cloned into the PCFD3 vector [21]. Transgenic gRNA flies were created by the Best Gene Company (Chino Hills, USA) using *y sc v P{nos-phiC31\int*.*NLS}X; P{CaryP}attP2* (BDSC 25710) flies. Transgenic lines were confirmed by sequencing (Biomedal, Armilla, Spain). Males carrying the sgRNA were crossed to *nos-Cas9* females, and the progeny was scored for the v + ch- eye marker. To identify CRISPR/Cas9-induced mutations, genomic DNA was isolated from flies and sequenced with the following primers (5′-3′):

*PVRAP* primer Forward: GTCCTGGTGGTGACTGGAAC;

*PVRAP* primer Reverse: AATCGCATAGCTGCCAACTT.

Two *PVRAP* null mutant alleles were generated, *PVRAP^1^* and *PVRAP^2^*. *PVRAP^1^* was a deletion of 2 base pairs in exon 1, which resulted in a frame-shift generating a stop codon after amino acid 29. *PVRAP^2^* carried an insertion of 8 base pairs, which resulted in a frame-shift generating a stop codon after amino acid 80.

### 2.3. Immunocytochemistry, In Situ Hybridization, Adult Wing Mounting and Imaging

Wing imaginal discs were stained using normal procedures and mounted in Vectashield (Vector Laboratories, Burlingame, CA, USA). We used the following primary antibodies: goat anti-GFP^FICT^ (Abcam, 1:500), rabbit anti-aPKC (Santa Cruz Biotechnology, 1:300), mouse anti-RFP (Proteintech, 1:500), rabbit anti-PH3 (EMD Millipore Corporation; 1:250), rabbit anti-caspase Dcp1 (Cell Signaling; 1:100) and rabbit anti-pJNK (Promega, 1:200). The secondary antibodies used were as follows: goat anti-mouse Alexa-488, Cy3 and Cy5 (Life Technologies, 1:200) goat anti-rabbit Cy3 and Cy5 (Life Technologies, 1:200) and goat anti-rat Cy3 (Life Technologies, 1:200). F-actin was visualized by means of Rhodamine Phalloidin (Molecular Probes, 1:50). DNA was marked with Hoechst (Molecular Probes, 1:1000).

Confocal images were acquired with a Leica SP5-MP-AOBS or a Zeiss LSM 880 microscope, equipped with a Plan-Apochromat 20× oil objective (NA 0.7), 40× oil objective (NA 1.4) and 63× oil objective (NA 1.4).

In situ hybridization was performed using standard procedures. A digoxygenin-UTP (Boerhringer-Mannheim, Mannheim, Germany)-labeled *PVRAP* anti-sense RNA probe was generated using the plasmid cDNA EST RE08107 (DGRC).

### 2.4. Quantification of Fluorescence Intensity

To quantify fluorescence intensity, fluorescent signaling was measured using the square tool in FIJI-Image J. Several confocal images per genotype were measured.

To calculate cell areas, the Huang threshold algorithm was applied to maximum projections of confocal sections. Cell volumes were estimated considering wing disc cells as truncated prisms and using the formula Volume = Height/3 (Basal Area + Apical Area + √ Basal Area × Apical Area).

To measure cell height, a vertical line was drawn from the apical to the basal surface in a region of interest of a wing disc stained for F-actin to visualize cell limits. The total length of the resulting line was measured using FIJI-ImageJ software.

Cell proliferation was quantified using the Trainable Weka 2D Segmentation plug-in, which transforms 8-bit images into a binary system. Dots of fluorescence intensity of wing discs stained with an anti-PH3 antibody were analyzed using the FIJI-Image J tool analyze particles.

Statistical comparisons were performed using the Welch test and chi-square tests.

## 3. Results

### 3.1. The Knockdown of PVRAP or by Enhances Ras^V12^ Hyperplastic Phenotype

Ectopic expression of activated *Ras* (*Ras^V12^*) in *Drosophila* wing discs leads to hyperplasia as a consequence of an increase in cell growth, accelerated G1-S transition and changes in cell shape [10,12] (Figure 1A,A’,B,B’). To test the role of *PVRAP* and *by* in *Ras^V12^*-mediated transformation, we reduced their levels in *Ras^V12^* wing disc tumor cells by expressing specific RNAis against these two genes. Ectopic expression of *Ras^V12^* in the dorsal compartment of wing discs, using the *apterous-Gal4* (*ap >* GFP; Ras^V12^, *n* = 50), leads to overgrowth of the tissue and formation of ectopic folds [10,12] (Figure 1A–B’,G). We found that although *PVRAP* or *by* RNAis had no obvious effect in control wing discs (*ap >* GFP; *PVRAP^RNAi^*, Figure 1C,C’,G, *n* = 40, and *ap >* GFP; *by^RNAi^*, Figure 1E, E’,G, *n* = 40), it enhanced the overgrowth and ectopic fold phenotype of *Ras^V12^* wing discs (*ap >* GFP; *Ras^V12^*; *PVRAP^RNAi^*, Figure 1D,D’,G, *n* = 38, and *ap >* GFP; *Ras^V12^*; *by^RNAi^*, Figure 1F,F’,G, *n* = 36).

### 3.2. Generation of PVRAP Mutant Alleles Using the CRISPR/Cas9 Technique

The *by* mRNA is expressed in wing discs and is highly enriched in the wing pouch [22]. To analyze PVRAP expression, we performed in situ hybridization in wing discs (see Materials and Methods). We found that the PVRAP transcript was abundantly expressed in the wing pouch of wild-type flies (Figure 2A).

While there are available null mutations for *by*, mutations in *PVRAP* have not yet been isolated. Thus, to better illustrate the role of *PVRAP* as a modulator of *Ras^V12^*-mediated hyperplasia, we used CRISPR/Cas9 to generate specific *PVRAP* alleles (Materials and Methods). The *PVRAP* gene encodes for only one transcript (*PVRAP*-RA, Flybase), whose transcription start site maps to the beginning of exon 1 (Figure 2B). We generated two *PVRAP* mutant alleles, *PVRAP^1^* and *PVRAP^2^,* which truncate 97% and 96% of the PVRAP protein, respectively, and can be considered therefore as null mutations by molecular criteria (Figure 2C). In addition, no *PVRAP* mRNA expression was detected in wing discs from these mutants (Figure 2B). As is the case for *by* [19,22], *PVRAP* mutant alleles were homozygous-viable. However, while *by* mutant flies show a fully penetrant wing blister phenotype [19,22], PVRAP mutant flies did not display any detectable morphological aberrations, implying that *PVRAP* is unessential for development in *Drosophila*.

To confirm the role of *PVRAP* and *by* as modulators of *Ras^V12^*-mediated tumorigenesis, we tested for synergetic interactions between *PVRAP* or *by* mutations and *Ras^V12^* in wing discs (Figure 3). We found that expression of *Ras^V12^* in the posterior compartment of *PVRAP* (*ap > Ras^V12^*; *PVRAP^1^*, *n* = 20), or *by* (*ap > Ras^V12^; by^33c^*, *n* = 16) mutant discs resulted in an enhancement of the folding phenotype, similar to that found in *Ras^V12^*; *PVRAP^RNAi^* and *Ras^V12^*; *by^RNAi^* wing discs (Figure 1 and Figure 3).

### 3.3. PVRAP and by Restrain Ras^V12^ Hyperplastic Phenotype by Regulating Cell Shape Changes and Growth

The formation of additional folds could be due to changes in cell polarity, proliferation, shape or a combination of some or all of these cellular properties. The role of human tensins in cell proliferation in normal and cancer cells is complex and tensin-type-specific. Thus, while knockdown of TNS1, 3 and 4 reduces proliferation in several normal and cancer cell lines, overexpression of TNS2 reduces the proliferation of several cancer cell lines (reviewed in [13]). Thus, we next decided to test whether the removal of *PVRAP* or *by* would affect the proliferative state of Ras^V12^ cells. Previous results have shown that overexpression of Ras^V12^ results in a reduction in the number of wing disc cells in mitosis, as revealed with an antibody against phosphorylated histone H3 (PH3) (Appendix A) [10,12,23]. Here, we found that while removal of *PVRAP* did not affect the proliferation of wing imaginal disc cells (Appendix A, *n* = 25), elimination of *by* led to an increase in cell proliferation (Appendix A, *n* = 20). In addition, elimination of either *PVRAP* (*ap > Ras^V12^*; *PVRAP^1^*, Appendix A, *n* = 13) or *by* (*ap > Ras^V12^; by^33c^*, Appendix A, *n* = 20) enhanced the proliferative capacity of Ras^V12^ wing imaginal discs.

Although ectopic *Ras^V12^* expression in wing disc cells alone does not affect cell polarity ([24] Appendix A), the elimination of polarity genes increases the *Ras^V12^*-mediated hyperplasia [25]. Thus, we analyzed if cell polarity was affected in *Ras^V12^*; *PVRAP1* or *Ras^V12^; by^33c^* disc cells. In order to do this, we analyzed the localization of the apical polarity marker atypical protein kinase C (aPKC) [26] in control and experimental conditions (Appendix A). We found that elimination of either *PVRAP* or *by* did not alter the localization of aPKC in normal (Appendix A) and *Ras^V12^* cells (Appendix A), suggesting that polarity was not affected.

To analyze possible cell shape changes, we visualized cell limits using Rhodamine-Phalloidin (Rh-Ph) that labels F-actin and therefore the cell cortex (Figure 4). The wing pouch cells in late third-instar control discs are columnar, with a mean height of 25 μm, an apical area of 8.93 μm^2^ and a basal area of 9.41 μm^2^ (Figure 4A–A’’’,G,H,I). In contrast, *Ras^V12^*-expressing cells have been found to be shorter and more cuboidal, with a mean height of 19 μm, an apical area of 10.72 μm^2^ and a basal area of 15.86 μm^2^ (Figure 4D–D’’’,G,H,I [12]. Considering disc cells as truncated prisms, this has been shown to result in an increase in cell volume [12]. This is in agreement with previous observations showing that *Ras^V12^* cells display increased cellular growth [10,23]. Here, we found that the elimination of either PVRAP or *by* in normal cells (*n* = 109 and 105, respectively) did not have any effect on cell size (Figure 4B–B’’’,C–C’’’,G,H,I). In contrast, the knockdown of *PVRAP* or *by* in *Ras^V12^* cells enhanced the expansion of the apical area around 30% and 27%, respectively (Figure 4E–E’,F–F’,G) but did not cause any effect on either the basal area (Figure 4E’’,F’’,H) or the height (Figure 4E’’’,F’’’,I). This, besides causing a change in cell shape, if we consider disc cells as truncated prisms, resulted in a further increase in cell volume of 15% and 13% upon removal of *PVRAP* or *by*, respectively.

Besides an increase in hyperplastic growth, the expression of *Ras^V12^* in wing disc cells has also been shown to promote the transition from G1 to the S phase [10]. Furthermore, the use of Fly-Fucci, which relies on fluorochrome-tagged degrons from cyclin B, degraded during mitosis, and E2F1, degraded at the onset of the S phase (Figure 5A), has confirmed the role of *Ras^V12^* in driving the transition from G1 to the S phase [27], showing that *Ras^V12^*-expressing wing discs exhibited an increase in the G2 population, 68.39% versus 42.05% in control wing discs, and a reduction in the G1 population, 22.64% vs. 33.61% in controls (Figure 5B,B’,C,C’, number of *Ras^V12^* cells = 10,965 cells, number of control cells = 35,768 cells). Using Fly-Fucci, we found that expression of an RNAi against either *PVRAP* or *by* did not affect progression through the cell cycle (Figure 5D,D’,F,F’). In addition, reducing the levels of *PVRAP* did not seem to substantially change the behavior of *Ras^V12^* cells, with populations of G2 and G1 of 64.75% and 26.92%, respectively (Figure 5E,E’, *n* = 27,211 cells). Unfortunately, for unknown reasons, flies carrying the Fly-Fucci transgenes and co-expressing *Ras^V12^* and an RNAi against *by* under the control of the *apGal4* driver were not viable. Thus, we could not assess the effect of the removal of *by* in the progression of the cell cycle of *Ras^V12^* cells.

Altogether, these results suggest that *PVRAP* and *by* modulate *Ras^V12^*-mediated tissue hyperplasia by enhancing cell shape changes and cellular growth. In view of the constraints imposed by the peripodial membrane, we suggest that the formation of extra folds could be explained by the cell shape changes and the increase in cell size.

### 3.4. Consequences of PVRAP or by Elimination in the Non-Autonomous Effects of Ras^V12^ Tumor Cells

Preceding observations have shown that JNK activity was elevated in wild-type cells surrounding *Ras^V12^*-expressing cells (Figure 6A,B,G; [12,28]). This has been shown to non-autonomously activate the JAK-STAT pathway in the tumor cells promoting their growth [28]. As we show here that elimination of either *PVRAP* or *by* enhanced the growth of *Ras^V12^* tumor cells, we tested whether the activity of the JNK pathway was affected in these tumor conditions.

Previous analysis has shown that *by* genetically interacts with the JNK pathway and that the activity of this pathway, measured by examining the extent of JNK phosphorylation using an anti-phosphospecific JNK antibody (pJNK), is dramatically increased or reduced upon *by* overexpression or downregulation, respectively [22]. However, here we found that pJNK levels did not change in either *PVRAP^1^
*(*n*= 17) or *by^33c^* (*n* = 17) mutant wing discs compared to controls (*n* = 14) (Figure 6A,C,E,G). In addition, we found that the levels of JNK activity in wild-type cells next to *Ras^V12^* tumor cells mutant for *PVRAP* (Figure 6D,G, *n* = 12) or *by* (Figure 6F,G, *n* = 17) were not significantly different from those found in wild-type cells adjacent to *Ras^V12^* (Figure 6B,G, *n* = 20) tumor cells. These results suggest that proteins of the tensin family modulate the growth of tumor cells independently of the JNK pathway.

Overexpression of activated Ras has also been shown to promote the death of nearby wild-type tissue in *Drosophila* imaginal tissues [12,23]. Consistent with this, an enrichment of apoptosis was noticed in wild-type (GFP-negative) ventral cells located at the D/V boundary in *ap >* GFP; *Ras^V12^* discs (*n* = 24) compared to controls (*n* = 23) (Appendix A; [12]). Here, we found that while removal of *by* did not affect apoptosis in wing imaginal discs (*n* = 19, Sup. Figure 3E,E’,G), elimination of *PVRAP* led to a general increase in cell death in the wing disc, with no clear concentration at the D/V boundary (*n* = 21, Appendix A). In addition, we found that elimination of either *PVRAP^1^* (*ap > Ras^V12^*; *PVRAP^1^*, Appendix A, *n* = 20) or *by^33c^* (*ap > Ras^V12^; by^33c^*, Appendix A, *n* = 20) enhanced the cell death of wild-type cells at the D/V boundary in *Ras^V12^*-expressing wing discs.

Apoptosis of wild-type cells near *Ras^V12^* cells has also been attributed to an enhancement in tissue compaction due to the overgrowth of mutant cells [29]. In fact, we have previously found that the wild-type ventral region of *ap >* GFP; *Ras^V12^* discs (*n* = 40) was more compressed than that of *ap >* GFP discs (*n* = 34) (Figure 3A’’,B’’, [12]. This *Ras^V12^* phenotype was further enhanced in *PVRAP* (*n* = 20) and *by* (*n* = 16) mutant backgrounds (Figure 3D’’,F’’).

Effector caspases are active in tumors, and this has been associated with metastasis [30]. In fact, caspase activity has been shown to induce the migration of transformed cells in wing imaginal discs [31]. In agreement with this, in a previous study, we reported the presence of *Ras^V12^* cells positive for Dcp1 in the ventral domain of *ap > GFP*; *Ras^V12^* [12], Appendix A). Tensins have been reported to play a role in cell migration and invasion. However, experiments in cell culture analyzing the roles of the different tensins in cell invasion have yielded contradictory results and appeared to be cell context-dependent (reviewed in [13]). Thus, we tested whether the elimination of either *PVRAP* or *by* would affect the migration of normal or *Ras^V12^* tumor cells. In order to do this, we analyzed the presence of GFP+ cells outside of the dorsal domain in control and experimental wing discs. We found that in either *ap >* GFP; *PVRAP^1^* (*n* = 25) or *ap > GFP; by* mutant wing discs (*n* = 28), the cells did not invade the ventral compartment (Appendix A). In addition, we found that the removal of either *PVRAP* (*n* = 36) or *by* (*n* = 36) did not affect the number or the migration distance of *Ras^V12^* GFP+ tumor cells in the ventral compartment (Appendix A).

### 3.5. Genetic Interactions between Proteins of the Tensin Family

The absence of a loss-of-function phenotype for *PVRAP* could be explained by compensation by other proteins of the tensin family performing similar functions, such as *by*. Thus, we examined whether *PVRAP* would genetically interact with *by*. As *PVRAP* and *by* are both on the third chromosome, we analyzed the effects of reducing the levels of both *PVRAP* and *by* by expressing a *PVRAP^RNAi^* in the posterior compartment of *by* mutant wing discs. While flies homozygous for *by^33c^* are viable and show blisters in the wing [19,22] (Figure 7B), flies heterozygous for *by^33c^* (*by^33c^*/+) or flies expressing an RNAi against *PVRAP* in the posterior compartment of control wing discs did not show any visible adult phenotype (*ap >* GFP; *PVRAP^RNAi^*, Figure 7C). In contrast, expression of *PVRAP^RNAi^* using the *apGal4* driver in a homozygous *by^33c^* genetic background (*by^33c^*; *ap >* GFP; *PVRAP^RNAi^*) was lethal, with a very small fraction of flies (2%, *n* = 100) reaching adulthood. These escapers showed strong defects in the wing and the notum and died soon after hatching (Figure 7D). In addition, expression of a *PVRAP^RNAi^* in the posterior compartment of the wing imaginal disc of heterozygous *by^33c^* resulted in a smaller scutellum and reduced number of bristles (*by^33c^*/+; *ap >* GFP; *PVRAP^RNAi^*; Figure 7E,F). Interestingly, this phenotype was also observed when an RNAi against *EGFRAP*, the ortholog of human TNS2 (Flybase), is expressed in the dorsal compartment in a heterozygous *by^33c^*/+ genetic background (*by^33c^*/+; *ap >* GFP; *EGFRAP^RNAi^*; Figure 7G).

Mammalian tensins are known to participate in integrin signaling [32]. Similarly, as mutations in *by* genetically interact with viable integrin alleles, the *Drosophila* tensin has also been proposed to functionally interact with integrins during wing development [22]. Here, we show that *by* also interacts with *PVRAP*. Thus, we next tested whether *PVRAP* would also interact with integrins. As mentioned above, the total elimination of *PVRAP* or its downregulation in the dorsal domain of wing imaginal discs (*ap > PVRAP^RNAi^*) on its own did not cause any visible phenotype in the adult (Figure 7C). Integrins are heterodimers composed of an α and a β subunit. Two integrins, which share the same β subunit, are expressed in the *Drosophila* wing disc, the αPS1 βPS (PS1) on the dorsal side and αPS2 βPS (PS2) in the ventral domain. Eliminating integrin activity in the wing imaginal disc results in blisters in the adult appendage (reviewed in [33,34]). Here, we found that the co-expression of a *mys^RNAi^* and a *PVRAP^RNAi^* in the dorsal region of wing discs (*ap > PVRAP^RNAi^*; *mys^RNAi^*) led to lethality. This result suggests that PVRAP also interacts with integrins.

## 4. Discussion

Cancer is a devastating disease that threatens human health worldwide [35]. One of the most frequently affected genes in cancer is the proto-oncogene Ras. In fact, mutations leading to its overactivation are present in ~30% of human cancers [36]. However, hyperactivation of Ras signaling alone is not sufficient to produce malignancy; additional mutations in other genes are required to drive Ras-dependent tumorigenesis (reviewed in [37]). Thus, identifying genes that modulate the oncogenic capacity of Ras is vital in our fight against cancer. The tensin family of focal adhesion proteins has emerged as a regulator of tumor progression in many cancer types [38]. However, the role of tensins in cancer is not fully established, since they can serve either as cancer-promoting or as cancer-inhibitory factors. Most experimental studies have mainly explored the positive or negative effects of tensins in in vitro cell culture models of different cancer cell lines (reviewed in [13]). Thus, a better understanding of the role of tensins in cancer development in the context of a whole organism is still missing. Here, we have used the *Drosophila* model to analyze the mechanism by which tensins modulate the progression of epithelial tumors. We show that *by*, the clearest homolog of human TNS4, and *PVRAP*, an ortholog of human TNS2 (FlyBase), act as tumor suppressors of Ras-mediated tumorigenesis, as their elimination enhances the overgrowth due to the overexpression of oncogenic Ras in wing disc epithelial cells. In addition, we find that tensins regulate tumor progression by restraining cell proliferation, cell cycle progression and cellular growth. Our results suggest that the role of tensins as cancer-inhibitory factors has been conserved across evolution and unravel possible mechanisms of action.

Tensins belong to the family of adhesion proteins that form focal adhesions and serve as a bridge between the extracellular matrix and the intracellular actin cytoskeleton. The mammalian tensin family comprises four members, which are multidomain proteins; each contains, on its N-terminal region, an actin-binding domain, which overlaps with a focal-adhesion-binding site and PTEN-like protein tyrosine phosphatase and C2 domains, and, on its C-terminal region, an Src homology 2 (SH2) domain and a phosphotyrosine binding domain. These domains allow tensins to transduce several signaling pathways, such as PI3/Akt and b-integrin/Fax pathways, regulating a variety of physiological processes, including cell proliferation, survival, adhesion, migration and mechanical sensing. Analysis of the role of mammalian tensins in mice has revealed that while individual tensins are not essential for embryonic or tissue development, they are required to maintain the structure and function of the kidney and heart and for regeneration processes (reviewed in [13]). As for their role in tumorigenesis, this appears to be quite controversial, and tensin- and cell-type-specific, as they act sometimes as tumor suppressors and other times as tumor promoters (reviewed in [39]). Unlike mammals, *Drosophila melanogaster* and *Caenorhabditis elegans* have been proposed to possess one tensin each. In addition, while the worm tensin is more similar to TNS1, TNS2 and TNS3 and contains all domains present in these tensins, the fly one (*by*) is more similar to TNS4 and contains only the SH2 and PTB domains [19,22,40]. As is the case in mammals, tensin knockout in flies and worms has no impact on development and survival, and the role of tensins in tumor progression in these model systems had not been studied [19,22,40]. Besides *by*, two other genes, *EGFRAP* and *PVRAP*, have SH2 domains that are similar to the ones present in mammalian tensins and have been suggested to be orthologs of human TNS2 and TNS4, respectively (FlyBase). Similar to *by*, the elimination of either *EGFRAP* [12] or *PVRAP* (this work) has no consequences for development and survival. In addition, the elimination of any of these three *Drosophila* tensins enhances the overgrowth due to the overexpression of oncogenic Ras ([12] and this work). These results suggest that in *Drosophila*, all tensins seem to behave similarly with respect to their ability to suppress tumor progression, at least in epithelial wing imaginal disc cells. In the future, it will be interesting to analyze their role in other cancer cell types, such as those produced in the gut endoderm or the brain. Finally, our results also suggest that, as could be the case in mammals, these *Drosophila* tensins may have redundant functions, since the downregulation of any one of them enhances the phenotype of eliminating each of them individually ([12] and this work). In the future, it would be interesting to further support this idea by generating flies double mutants for any two of the tensin genes.

As mentioned above, and similar to what we have found in *Drosophila*, mammalian tensins are not cancer-driver molecules. Instead, they seem to act as modulators of tumor progression, and they do so by regulating various cellular events, including cell polarization, proliferation, apoptosis and migration (reviewed in [13]; Pryczynicz, 2020 #1758). TNS1, TNS3 and TNS4 knockdown reduces the proliferation of several cancer cell lines, such as colon cancer and acute myeloid leukemia cell lines [41,42,43]. In contrast, TNS2 overexpression reduces cell proliferation and survival of some cervical and lung cancer cells [44]. In *Drosophila*, the overexpression of oncogenic *Ras^V12^* in the wing disc results in a reduction in the number of cells in mitosis [10,12,23]. In contrast to our previous results showing that *EGFRAP* does not affect the proliferation of *Ras^V12^* in wing discs [12], here we find that elimination of either *PVRAP* or *by* slightly, but significantly, increases the number of *Ras^V12^* cells undergoing mitosis, suggesting that as is the case for the mammalian tensins, the different *Drosophila* tensins may regulate the behavior of tumor cells in a tensin-type-specific manner. However, even though the number of cells in mitosis in tumorigenic wing discs mutant for either *PVRAP* or *by* is higher than that found in tumorigenic wing discs, it is still lower than that found in controls. Thus, this increase on its own cannot account for the increase in tissue overgrowth found in *Ras^V12^* wing discs mutant for either *PVRAP* or *by*, suggesting that these two tensins modulate *Ras^V12^*-dependent tumorigenesis by regulating additional cellular events rather than just cell proliferation. Previous analysis has demonstrated that *Ras^V12^* cells show increased cellular growth [10,12,23]. Here, we find that the elimination of either *PVRAP* or *by* in *Ras^V12^* cells results in a cell shape change, which leads to an increase in cell volume. This result unravels a new mechanism by which tensins could modulate tumor progression, the regulation of cell shape and growth.

Analysis of the roles of tensins in cell migration and invasion has shown that depending on the cellular context, tensins can either promote or inhibit cell migration. Thus, while the knockdown of TNS1 reduces the migration of mouse fibroblasts and endothelial cells, the overexpression of TNS1 or TNS2 promotes the migration of human embryonic kidney cells [45,46]. In addition, downregulation of TNS1, TNS2 or TNS3 reduces the invasiveness of ovarian and breast cancer cell lines by impairing integrin internalization and focal adhesion turnover [47,48,49]. However, here we show that the elimination of either *PVRAP* or *by* does not affect the migration of normal or *Ras^V12^* tumor epithelial wing disc cells. One possible explanation for this result is that the function of tensins in *Drosophila* might be different from that in mammals. In fact, while tensins have been shown to affect integrin internalization in mammalian cells (see above), integrin localization is not affected in *by* mutant wing discs [19]. Furthermore, *Drosophila by* only affects a subset of integrin-mediated adhesion [19]. An alternative explanation is redundancy among the *Drosophila* tensin family of proteins. In the future, it will be interesting to analyze the consequences of simultaneously reducing *PVRAP* and *by* in normal and *Ras^V12^* tumor wing disc cells in cell migration. In addition, and in order to better understand the contribution of tensins to *Ras^V12^*-induced tumorigenesis, it will be interesting to analyze the effects of overexpressing *PVRAP* and *by* on the *Ras^V12^* phenotype.

Finally, our results also show that fly tensins do not seem to regulate the polarization or survival of tumor cells. However, they seem to influence the ability of tumor cells to induce the apoptosis of nearby wild-type tissue. As the removal of either *PVRAP* or *by* enhances the formation of extra folds due to the overexpression of *Ras^V12^*, we propose that the increase in cell death in nearby wild-type tissue could be a direct consequence of greater compaction due to the higher growth of *Ras^V12^* cells in *PVRAP* or *by* mutant backgrounds.

Through their different domains, mammalian tensins can bind pathway signaling molecules, including b1-integrin, PI3K/Akt/mTOR, FAK, Rho GAP, p130Cas, TGF-b and the Ras/Raf pathways, thus regulating a myriad of different cellular responses in normal cells (reviewed in [13]; Pryczynicz, 2020 #1758; Mouneimne, 2007 #1464). Even though most studies analyzing the role of mammalian tensins in cancer have been dedicated to assessing the expression of tensins in different cancer types, there have been studies attempting to identify the role of tensins in carcinogenesis. Thus, TNS1 has been proposed to regulate tumor cell proliferation affecting Rho GAP through regulation of the hippo signaling pathway (reviewed in [38]). Other studies indicated that TNS4 could promote cancer progression by regulating the Akt/GSK-3b and TGF-b1 signaling pathways [50,51]. In addition, a reciprocal TNS3-TNS4 switch regulates the invasive capacity of breast tumors downstream of the EGFR via direct interaction with b1 integrins [52]. In *Drosophila*, *by* interacts with integrins and the JNK pathways [19,22], while PVRAP physically interacts with PVR [53] and EGFRAP interacts and regulates the EGFR pathway [12], interactions that regulate adhesion and fate in normal cells. In *Drosophila* wing disc tumor cells, we have recently shown that EGFRAP restrain the oncogenic capacity of EGFR/Ras hyperactivation [12]. Here, we show that *by* also acts as a tumor suppressor in Ras-mediated oncogenesis. As *by* has been shown to interact with integrins [19,22] and we have recently shown that b1 integrins also behave as suppressors of Ras^V12^-dependent tumorigenesis in *Drosophila* wing discs [17], we propose that *by* may act as a tumor suppressor via its ability to bind and regulate integrins. This suggests that the function of tensins as tumor modulators via regulating integrin function might have been conserved throughout evolution. Finally, overactivation of PVR has also been shown to produce overgrowth of the *Drosophila* wing disc [54]. As *PVRAP* interacts with PVR [53], we propose that *PVRAP* could act as a tumor suppressor by regulating PVR activity, similar to the relationship between EGFRAP and EGFR. Altogether, these results suggest that, similar to what happens with mammalian tensins, the *Drosophila* tensin orthologs could modulate tumorigenesis by regulating different signaling pathways (Figure 8). Finally, our results showing that downregulation of *PVRAP* and *EGFRAP* led to defects consistent with downregulation of integrin function [12] suggest the existence of cross-talks between the different *Dosophila* tensins and the pathways they can modulate.

Even though tensins have been widely implicated in different types of cancers, to date, there is no clinical trial targeting them, as their impact on carcinogenesis is not fully established. Our results demonstrate that *Drosophila* tensins act as suppressors of Ras^V12^ tumor progression in wing disc epithelial cells. We can now use the advantages of the *Drosophila* system to increase our understanding of the mechanisms by which tensins modulate carcinogenesis and to identify new therapeutic drugs targeting malignancy, a top priority in cancer research.

## Figures and Tables

**Figure 1 genes-14-01502-f001:**
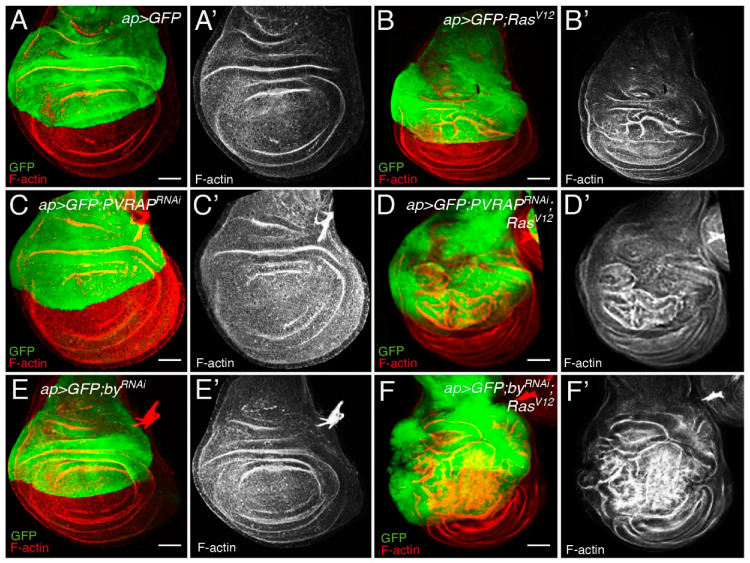
*by* and *PVRAP* knockdown enhances *Ras^V12^* hyperplasia in *Drosophila* wing discs. (**A**–**F’**) Maximal projection of confocal views of 3rd-instar larvae wing discs, expressing GFP (green) and the indicated UAS transgenes under the control of apterous Gal4 (*apGal4*) driver, stained with anti-GFP (green in (**A**–**F**)) and Rhodamine Phalloidin (RhPh) to detect F-actin (red in (**A**–**F**) and white in (**A’**–**F’**)). Scale bars 50 μm.

**Figure 2 genes-14-01502-f002:**
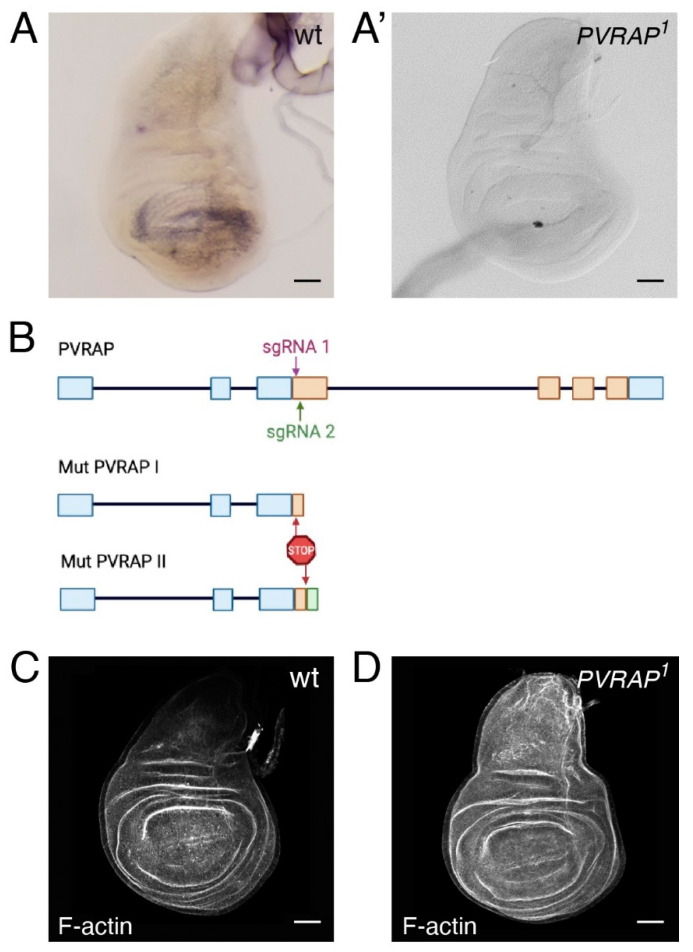
Generation of *PVRAP* mutant alleles by CRISPR-Cas9. (**A**,**A’**) In situ hybridization of 3rd-instar wild-type (**A**) and *PVRAP* mutant (**A’**) wing discs, using a probe for the *PVRAP* mRNA. (**B**) Schematic representation of the *PVRAP* locus, *PVRAP* mutants generated and sgRNAs used for the generation of the mutants (green and purple). (**C**,**D**) Maximal projections of confocal images of third-instar wing discs of the indicated genotypes stained with RhPh to detect F-actin (grey). Scale bars 50 μm.

**Figure 3 genes-14-01502-f003:**
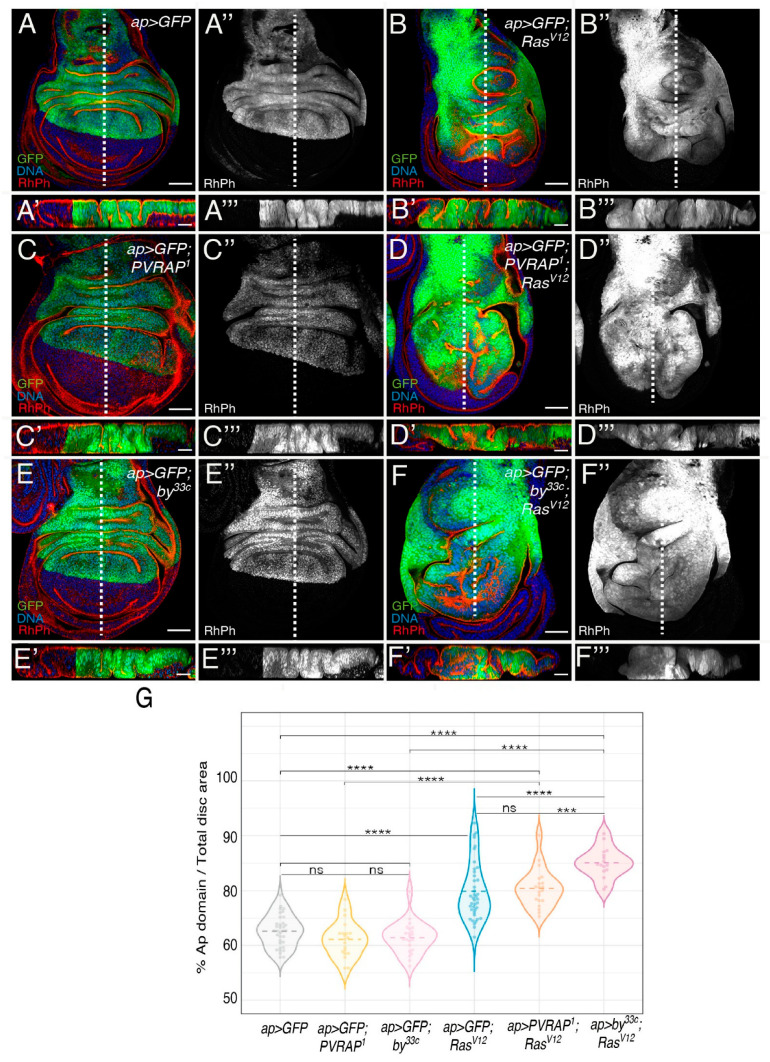
Hyperplasia due to *Ras^V12^* overexpression in wing discs is enhanced in *by* and *PVRAP* mutant backgrounds. (**A**–**F**) Confocal views of 3rd-instar larvae wing discs, expressing GFP (green), the indicated UAS transgenes driven by *apGal4*, in wild-type (**A**) and mutant genotypes (**B**–**F’’**), stained with anti-GFP (green in **A**,**A’**, **B**,**B’**, **C**,**C’**, **D**,**D’ E**,**E’ F**,**F’** and white in **A’’**,**A’’’**, **B’’**,**B’’’**, **C’’**,**C’’’**, **D’’**,**D’’’**, **E’’**,**E’’’**, **F’’**,**F’’’**), RhPh to detect F-actin (red) (**A’’**–**F’’’**) and Hoechst (DNA, blue). (**A’**, **A’’’**,**F’**,**F’’**) Confocal sections of wing discs of the specified genotypes along the white dotted lines shown in (**A**–**F**), respectively. (**G**) Violin plot of the percentage of GFP area per disc of the indicated genotypes. The statistical significance of differences was assessed with a Welch test, ****, *** *p* value < 0.0001 and <0.001, respectively. Scale bars 50 μm (**A**–**F**,**A’**–**F’’**) and 20 μm (**A’**–**F’**,**A’”**–**F’”**).

**Figure 4 genes-14-01502-f004:**
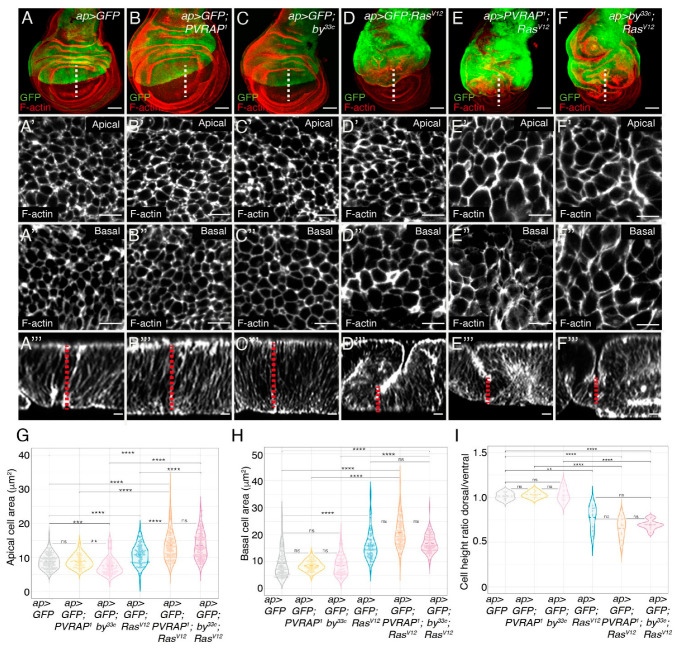
*Ras^V12^*-dependent cell shape changes and growth of *Drosophila* wing discs are increased in *by* and *PVRAP* mutant backgrounds. (**A**–**F**) Maximal projections of confocal images of wing imaginal discs from third-instar larvae of the indicated genotypes expressing GFP (green) and the designated UAS transgenes under the control of *apGal4*, stained with anti-GFP (green) and RhPh to detect F-actin (red in **A**–**F**, white in **A’**–**F’’’**). (**A’**–**F’**) Apical and (**A”**–**F”**) basal surface views of the indicated genotypes. (**A”’**–**F”’**) Confocal *xz* sections along the white dotted lines of wing discs shown in (**A**–**F**). The apical side of wing discs is at the top. The red dotted lines indicate cell height. (**G**–**I**) Violin plots of the apical (**G**) and basal (**H**) cell areas and cell height (I) of the indicated genotypes. The statistical significance of differences was assessed with a Welch test; ****, *** and ** indicate *p* values < 0.0001, <0.001 and <0.01, respectively. Scale bars 50 μm (**A**–**F**), 5 μm (**A’**–**F’**, **A”**–**F”**) and 10 μm (**A’”**–**F’”**).

**Figure 5 genes-14-01502-f005:**
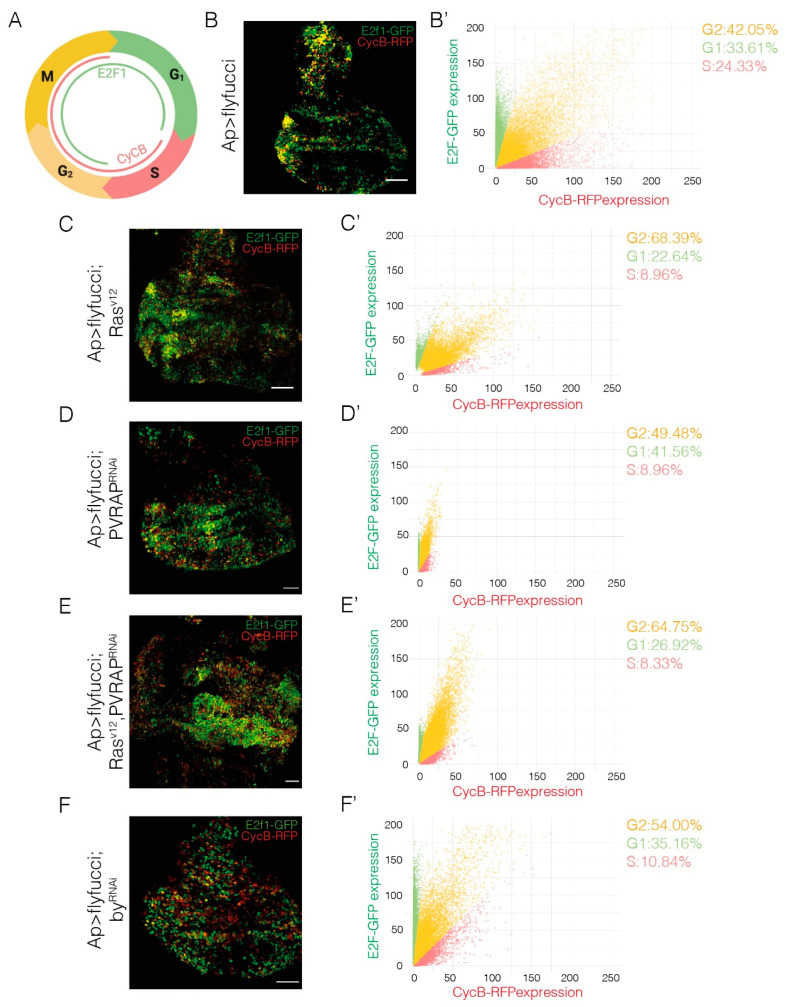
Elimination of *by* or *PVRAP* enhances the changes in cell cycle progression due to *Ras^V12^* overexpression in wing discs. (**A**) Scheme showing the expression of CycB-GFP and E2F1-RFP during the cell cycle. (**B**–**F**) Maximal projection of confocal images of 3rd-instar wing imaginal discs of the designated genotypes expressing the indicated UAS transgenes under the control of *apGal4*-*flyfucci*, stained for anti-GFP (green) and anti-RFP (red). (**B’**,**C’**,**D’**,**E’**,**F’**) Scatter plots representing the fluorescence intensity of both proteins in each cell. Scale bars 50 μm (**B**–**F**).

**Figure 6 genes-14-01502-f006:**
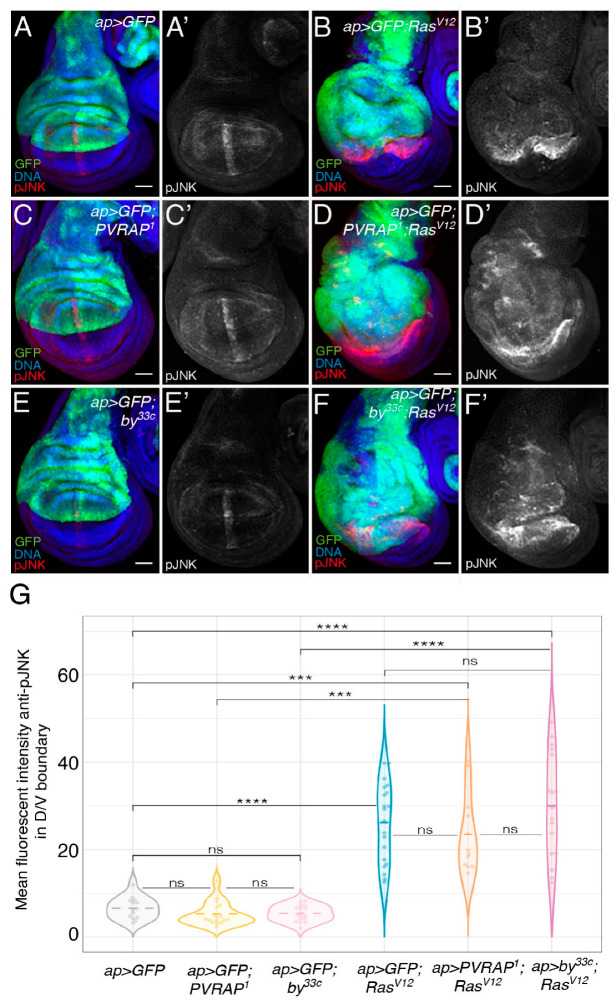
*by* and *PVRAP* do not affect the ability of *Ras^V12^* tumor cells to induce JNK activation in nearby wild-type tissue. (**A**–**F**) Maximal projections of confocal views of 3rd-instar wing discs expressing the indicated UAS transgenes under the control of *apGal4*, stained with anti-GFP (green), anti-pJNK (red in (**A**–**F**), white in (**A’**,**B’**,**C’**,**D’**,**E’**,**F’**)) and Hoechst (DNA, blue). (**G**) Violin plots of mean fluorescent pJNK intensity in the dorsal–ventral boundary of wing discs of the designated genotypes. The statistical significance of differences was assessed with a Welch test, **** and *** indicate *p* values < 0.0001 and <0.001, respectively. Scale bars, 50 μm (**A**–**F’**).

**Figure 7 genes-14-01502-f007:**
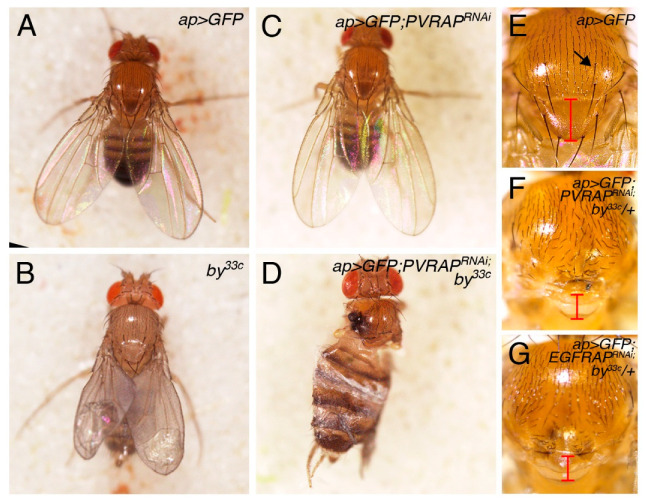
*PVRAP* downregulation enhances *by* loss-of-function phenotypes. (**A**–**D**) Dorsal views of adult *Drosophila* flies expressing the indicated UAS transgenes under the control of *apGal4*. (**E**–**G**) Dorsal views of the *Drosophila* notum of the specified genotypes.

**Figure 8 genes-14-01502-f008:**
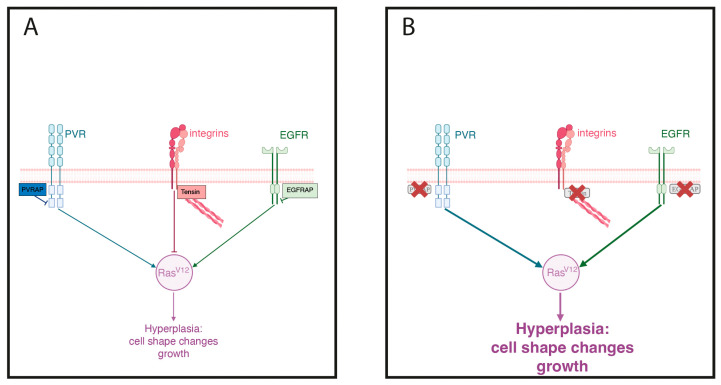
Model of *PVRAP* and *by* function as a modulator of Ras^V12^-dependent tissue hyperplasia. (**A**) Schematic drawing depicting the mechanisms by which *PVRAP*, *by* and *EGFRAP* could restrain Ras activity in Ras^V12^-dependent oncogenic cells. (**B**) Downregulation of any of these three tensin genes releases the restraint, leading to a further enhancement of Ras activation and tissue hyperplasia.

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
