# Peer review of "Drosophila* as Model System to Study Ras-Mediated Oncogenesis: The Case of the Tensin Family of Proteins"

_genes, 2023, doi:10.3390/genes14071502_

Round 1

Reviewer 1 Report

The manuscript of Millán et al is an excellent written manuscript and provides novel information on how two tensin Drosophila homologs, blistery (by) and PVRAP limits the RasV12 oncogenic capacity, by regulating cell shape and growth in the wing epithelium. Previous work in the same lab provided evidence that downregulation of PVRAP enhanced the RasV12 overgrowth phenotype in the wing tissue. To test further whether the overexpression of the oncogenic form of RasV12 synergized with PVRAP, the authors generated new loss-of-function alleles for PVRAP, which is an important genetic tool. Like by, flies without PVRAP are viable. The hyperplasia induced phenotype caused by the ectopic expression of RasV12 was further enhanced by the removal of either by or PVRAP.

The authors have performed a systematic and thorough analysis at the cellular level provided convincing data accompanied with the appropriate quantification analysis of various cellular parameters (e.g. surface area in basal and apical area, etc). The authors found that RasV12-dependent cell shape changes and growth of wing discs is increased upon removal of either blistery (by) or PVRAP. The authors used the Fly-Fucci system- an elegant genetic tool- and identified no implication of PVRAP in the ability of RasV12 to drive transition of cells from G1 to S phase. To further delineate the mechanism underlying in the non-autonomous effects of  RasV12, analyzed the activation of the JNK pathway, and found no implication of either by or PVRAP in the ectopic elevation of the active JNK in wild type cells surrounding the RasV12 expressing cells. However, the authors identified that both by and PVRAP enhanced the apoptosis-induced phenotype in wild-type cells near RasV12 expressing cells, although the responsible mechanism remains elusive. The authors were further addressed the genetic interactions between blistery (by) and PVRAP. Interestingly, the authors provided evidence that PVRAP does not play a role in the integrin-mediated adhesion in the wing epithelium, suggesting that blistery (by) and PVRAP differ in their functional requirement as modulators of integrin function. However, the authors suggested that PVRAP and integrins (mys) interact, because simultaneous knock-down of both genes in the dorsal region of wing discs led to lethality.

Points to address by the authors

1. The results and conclusions of the authors for the genetic interactions between blistery and PVRAP and RasV12 will be further corroborated if the authors prove further whether the levels of either blistery or PVRAP in the wing have a direct impact on the  RasV12- induced phenotype. Therefore, the authors together with the expression of  RasV12 in the wing epithelium should express a UAS construct of either blistery or PVRAP and assess whether the RasV12-induced phenotype is ameliorated. If the authors are unable to perform the suggested experiment (due to lack of revision time or availability of the appropriate fly stocks), they can mention the importance of the suggested experiment in the Discussion section.

2. The authors write in their manuscript that both blistery or PVRAP “are both on the second chromosome” (line 364). However, both genes are on the third chromosome: blistery (bl) or CG9379 is on the 3R (85D19-85D22), while the gene PVRAP or CG32406 is on 3L (65A2-A3). The authors should correct this misleading information.

3. The observed synthetic lethality of by33c; ap>GFP; PVRAPRNAi should be further tested by generating double homozygous mutants for both by33c and PVRAP1. If the authors are unable to perform the suggested experiment (due to lack of revision time or availability of the appropriate fly stocks), they can mention the importance of the suggested experiment in the Discussion section.

4. It would be informative for the broader audience if the authors provide a cartoon of their proposed model for the reported synergistic interactions of RasV12 and blistery (by) and PVRAP.

Author Response

We would like to take this opportunity to thank  Reviewer 1 for useful comments on the manuscript. This reviewer has raised several points, all of which have been addressed in the revised version of the manuscript. Here is our response to the reviewer´s comments:

  1. The results and conclusions of the authors for the genetic interactions between blistery and PVRAP and RasV12 will be further corroborated if the authors prove further whether the levels of either blistery or PVRAP in the wing have a direct impact on the RasV12- induced phenotype. Therefore, the authors together with the expression of RasV12 in the wing epithelium should express a UAS construct of either blistery or PVRAP and assess whether the RasV12-induced phenotype is ameliorated. If the authors are unable to perform the suggested experiment (due to lack of revision time or availability of the appropriate fly stocks), they can mention the importance of the suggested experiment in the Discussion section.

We agree with the reviewer that in order to better understand the contribution of tensins to RasV12-induced tumorigenesis, it will be interesting to analyse the consequences of overexpressing PVRAP and by on the RasV12 phenotype. Unfortunately, there are not reagents available to do this experiment within the frame revision time. Thus, following the reviewer´s advice, we have commented in the Discussion the appropriateness of carrying out this experiment.

  1. The authors write in their manuscript that both blistery or PVRAP “are both on the second chromosome” (line 364). However, both genes are on the third chromosome: blistery (bl) or CG9379 is on the 3R (85D19-85D22), while the gene PVRAP or CG32406 is on 3L (65A2-A3). The authors should correct this misleading information.

This has now been corrected.

  1. The observed synthetic lethality of by33c; ap>GFP; PVRAPRNAi should be further tested by generating double homozygous mutants for both by33c and PVRAP1. If the authors are unable to perform the suggested experiment (due to lack of revision time or availability of the appropriate fly stocks), they can mention the importance of the suggested experiment in the Discussion section.

We agree with the reviewer in that this is an important experiment. However, as both by and PVRAP genes are on the 2nd chromosome, generating double mutants will require recombining chromosomes carrying mutations in each of these genes. This could take a few months and unfortunately, we do not have so much revision time. Thus, following the reviewer´s advice, we have mentioned in the Discussion that generating these flies in the future will support possible redundancy between these genes and explain the synthetic lethality of by33c; ap>GFP; PVRAPRNAi.

  1. It would be informative for the broader audience if the authors provide a cartoon of their proposed model for the reported synergistic interactions of RasV12 and blistery (by) and PVRAP.

We have added a new Figure, Fig. 8, with a model for the reported synergistic interactions of RasV12 and blistery (by) and PVRAP.

Reviewer 2 Report

In the present work the authors analyzed the participation of two new factors of the tensin family in the proliferation of cells with Ras mutation. Using the Drosophila model, the authors demonstrated the role of mutations of tensin homologs in the hyperplasia of imaging disc cells in RasV12 mutants. Alterations in cell shape and growth were found. The genes act as suppressors of Ras-induced cellular transformation.

The work is well designed. The experimental work was carried out at a high level. An interesting discussion of the data is given.

In general, the work leaves the best impression and can be recommended for publication.

It can be suggested to the authors to read the text carefully and check for typos. I have noticed the following:

Line 111 - the number of nucleotides is not given

Line 191 - typo

Line 292 - no parenthesis

Lines 235, 237 - unclear unit of measurement (meter?)

Lines 444, 391 - typo in reference

Lines 542-563 - should be deleted

Author Response

We are happy to hear that Reviewer 2 has found our work interersting and well carried out. In fact, this reviewer has only suggested correction of several typos, which we have fixed in the revised version.